# Testing the mutant selection window hypothesis with meropenem: *In vitro* model study with OXA-48-producing *Klebsiella pneumoniae*

**Kamilla N. Alieva**[1], **Maria V. Golikova**[1]*, **Svetlana A. Dovzhenko**[1], **Mikhail B. Kobrin**[1], **Elena N. Strukova**[1], **Vladimir A. Ageevets**[2], **Alisa A. Avdeeva**[2], **Ofeliia S. Sulian**[2], **Sergey V. Sidorenko**[2,3], **Stephen H. Zinner**[4]

**1** Department of Pharmacokinetics & Pharmacodynamics, Gause Institute of New Antibiotics, Moscow, Russia, **2** Pediatric Research and Clinical Center for Infectious Diseases, St. Petersburg, Russia, **3** North-Western State Medical University named after I. I. Mechnikov of the Ministry of Health of the Russian Federation, St. Petersburg, Russia, **4** Department of Medicine, Harvard Medical School, Mount Auburn Hospital, Cambridge, MA, United States of America

\* golikovaka@gmail.com

**Data Availability Statement:** All relevant data are within the paper and its Supporting Information files.

## Abstract

OXA-48 carbapenemases are frequently expressed by *Klebsiella pneumoniae* clinical isolates; they decrease the effectiveness of carbapenem therapy, particularly with meropenem. Among these isolates, meropenem-susceptible carbapenemase-producers may show decreased meropenem effectiveness. However, the probability of the emergence of resistance in susceptible carbapenemase-producing isolates and its dependence on specific *K. pneumoniae* meropenem MICs is not completely known. It is also not completely clear what resistance patterns will be exhibited by these bacteria exposed to meropenem, if they would follow the patterns of non-beta-lactamase-producing bacteria and other than beta-lactams antibiotics. These issues might be clarified if patterns of meropenem resistance related to the mutant selection window (MSW) hypothesis. To test the applicability of the MSW hypothesis to meropenem, OXA-48-carbapenemase-producing *K. pneumoniae* clinical isolates with MICs in a 64-fold range (from susceptible to resistant) were exposed to meropenem in a hollow-fiber infection model; epithelial lining fluid meropenem pharmacokinetics were simulated following administration of 2 grams every 8 hours in a 3-hour infusion. Strong bell-shaped relationships between the meropenem daily dose infused to the model as related to the specific isolate MIC and both the antimicrobial effect and the emergence of resistance were observed. The applicability of the MSW hypothesis to meropenem and carbapenemase producing *K. pneumoniae* was confirmed. Low meropenem efficacy indicates very careful prescribing of meropenem to treat *K. pneumoniae* infections when the causative isolate is confirmed as an OXA-48-carbapenemase producer.

**Funding:** Russian Science Foundation (grant number 21-74-10090) The funders had no role in study design, data collection and analysis, decision to publish, or preparation of the manuscript.

**Competing interests:** The authors have declared that no competing interests exist.

## Introduction

Life-threatening carbapenemase-producing Gram-negative bacteria are widely distributed globally and pose a serious threat to public health, causing difficult-to-treat hospital-acquired infections that are associated with high mortality rates, especially in immunocompromised and critically ill patients. [1–4]. *K. pneumoniae* clinical isolates are able to produce a wide range of carbapenemases that enable effective degradation of beta-lactam antibiotics. Among them, OXA-48 carbapenemases are most frequently expressed by *Klebsiella pneumoniae* clinical isolates [5–9] and decrease the effectiveness of beta-lactam antimicrobial therapy. Over the past few decades, meropenem has been extensively used to treat Gram-negative infections; its efficacy has steadily declined due to antimicrobial resistance, primarily mediated by bacterial carbapenemases. A particular pattern of meropenem resistance may be exhibited by organisms whose main resistance mechanism is the production of carbapenemases. Nevertheless, it is not fully understood what fundamental resistance patterns will be exhibited by these bacteria exposed to meropenem, if they would follow the patterns of other than beta-lactams antibiotics or not. To address these issues we tested the well-known mutant selection window hypothesis with meropenem. The mutant selection window (MSW) hypothesis is applicable [10] to explain resistance patterns of several classes of antibiotics [11–17]; however, only a few such studies have been reported with beta-lactams [12, 17].

To determine meropenem "concentration-resistance" relationships, its ability to restrict the development of resistance throughout a treatment course and to explore the applicability of the MSW hypothesis to meropenem, clinical OXA-48-carbapenemase-producing *K. pneumoniae* isolates with MICs that varied over a 64-fold range were exposed to meropenem in a hollow-fiber infection model (HFIM). In this study, epithelial lining fluid (ELF) meropenem pharmacokinetics were simulated following administration of 2 grams every 8 hours in a 3-hour infusion [18]. Several clinical studies have shown that prolonged meropenem infusion at this dose is associated with clinical success and/or reduced mortality rates compared to standard bolus injection or a 30-minute infusion [19–21]. The HFIM is an effective tool for studying antimicrobial efficacy; it enables the simulation of human antibiotic pharmacokinetics and the evaluation of the pharmacodynamics of clinical antibiotic courses [22–24].

## Materials and methods

Methods used to produce the dataset are shown in the flowchart (**Fig 1**).

### Antimicrobial agent, bacterial isolates and susceptibility testing

Meropenem powder was purchased from Sigma-Aldrich (St. Louis, MO). Clinical specimens from ICU patients in hospitals located in Moscow and Saint Petersburg (Russia) were collected from 2015 to 2019. The sources of the isolates were: phlegmon, blood, wound and BAL. *K. pneumoniae* strains were isolated and identified using standard microbiological and biochemical assays. The respective *K. pneumoniae* collection was stored at -80˚C. Carbapenemase production by *K. pneumoniae* isolates was detected by PCR. Five OXA-48-positive clinical isolates with meropenem MICs ranging from 0.5 to 32 mg/L were selected for the study: 1128 (meropenem MIC = 0.5 mg/L), 1456 (meropenem MIC = 2 mg/L), 1170 (meropenem MIC = 4 mg/L), 202 (meropenem MIC = 16 mg/L) and 75 (meropenem MIC = 32 mg/L). MICs were determined by the broth microdilution technique at an inoculum size of $5\times10^5$ CFU/mL [25] with cation-supplemented Mueller-Hinton broth (CSMHB) (Becton Dickinson, USA).

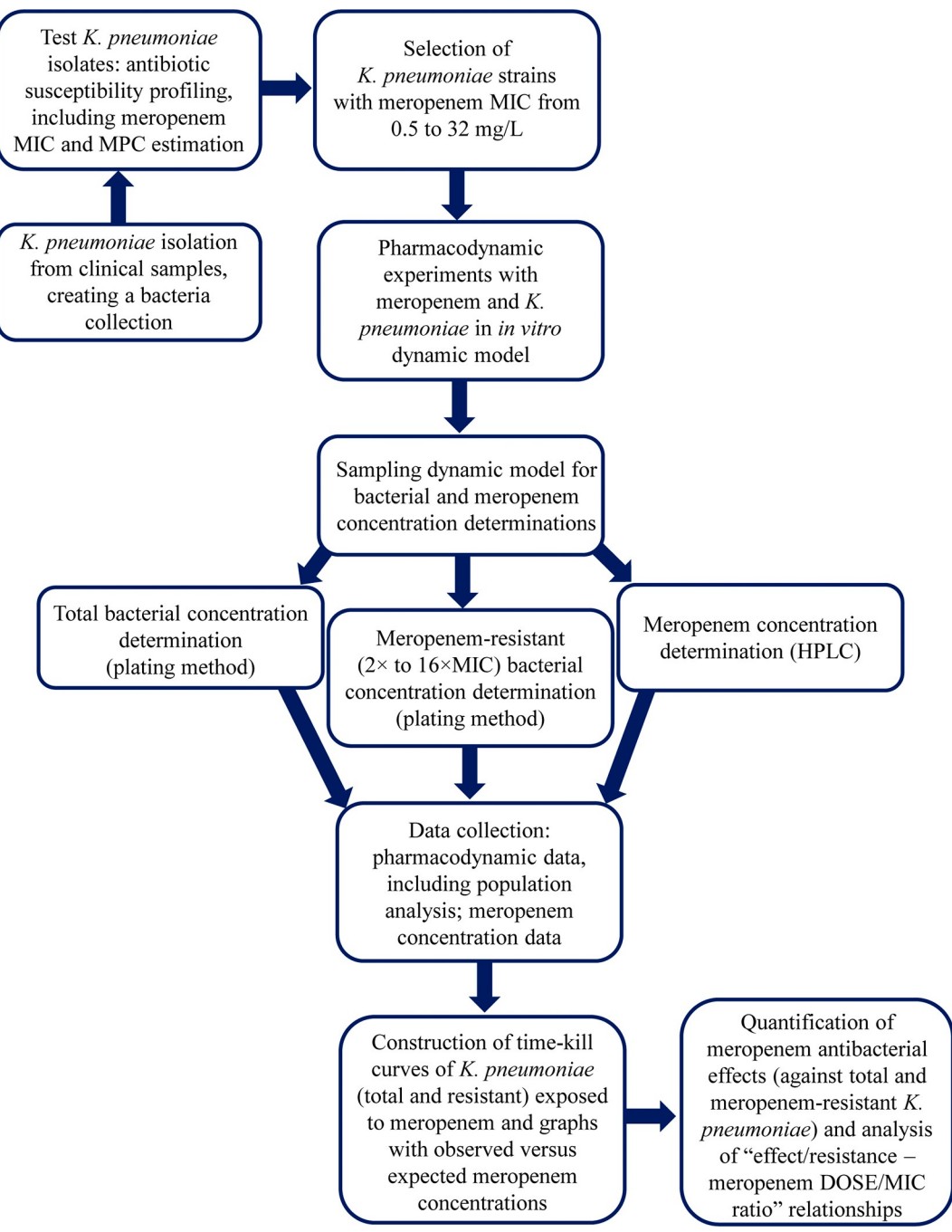

**Fig 1. Flowchart of the materials and methods section.**

### Mutant prevention concentration (MPC) determinations

The MPCs [26] for each of *K. pneumoniae* isolate were determined as described elsewhere [27]. Briefly, the tested microorganisms were cultured in CSMHB and incubated for 24 h. Then, the suspension was centrifuged (4000 × g for 10 min) and re-suspended in CSMHB to yield a concentration of $10^{10}$ CFU/mL. A series of agar plates (Mueller-Hinton agar, Becton Dickinson, USA) containing meropenem concentrations ranging from 0.5 to 2048 mg/L were

then inoculated with a suspension of *K. pneumoniae*, incubated for 24–48 h at 37˚C and screened visually for growth. The MPC was taken as the lowest meropenem concentration that completely inhibited growth. The lower limit of detection was 10 CFU/mL (equivalent to at least one colony per plate). The MPCs for *K. pneumoniae* 1128, 1456, 1170, 202 and 75 were estimated as 32, 192, 128, 96 and 64 mg/L, respectively.

### *In vitro* dynamic model and operational procedure used in the pharmacodynamic experiments

The HFIM was used to evaluate the meropenem pharmacodynamics. It was also used in growth control experiments. The studies were performed using hollow-fiber bioreactor (Fiber-Cell Systems, cellulose cartridge C3001) that represented the infection site.

The operational procedure is described in detail elsewhere [28]. Briefly, antibiotic dosing and sampling were processed automatically, using computer-assisted controls [GILSON 402 Syringe Pump, France (dosing); flow selectors of Valco Instruments Company Inc., USA (sampling)]. The system was filled with sterile CSMHB and placed in an incubator at 37˚C. The inoculum of an 18 h culture of *K. pneumoniae* was injected into the hollow-fiber bioreactor to produce a bacterial concentration of $10^8$ CFU/mL. After a 2 h incubation samples were obtained to determine the starting bacterial concentration; then, the infusion of CSMHB with antibiotic was started. Pharmacokinetic simulations were performed using peristaltic pumps (Cole-Parmer, Masterflex L/S, USA). The duration of each experiment was 120 h. To verify the reliability of pharmacokinetic simulations, throughout each experiment the bioreactor was sampled immediately after the end of infusion (3 hour) and at the 7th hour of the dosing interval. Meropenem concentrations were determined by HPLC.

### Antibiotic dosing regimens and simulated pharmacokinetic profiles

Meropenem treatment mimicked the therapeutic dosing regimen: 2 grams administered every 8 h, as a 3-h intravenous infusion. A mono-exponential profile in human epithelial lining fluid (ELF) after thrice-daily dosing of meropenem with a half-life ($t_{1/2}$) of 1.4 h was simulated for 5 consecutive days with a total of 15 infusions. The steady-state pharmacokinetic parameter values were as follows: $C_{MAX}$ = 32.4 mg/L, 24-h area under the concentration-time curve (AUC) = 375 (mg×h)/L [18]. Before all pharmacodynamic simulations to validate the system works properly, *in vitro* pharmacokinetic experiments in CSMHB without bacteria were conducted. For "dose–response" relationships the total daily doses of meropenem (in mg) that were infused to the system to simulate the desired meropenem pharmacokinetic profiles were calculated. Based on desired meropenem pharmacokinetic profile the $T_{MSW}$ (time in % of dosing interval, when targeted (not observed) meropenem concentration are inside the MSW [10]) index was calculated.

### Determination of meropenem concentrations by HPLC

**Chromatography conditions.** Isocratic separation was performed at 30˚C on a column Luna C18 (2) (250 mm × 4.6 mm, particle size 5 microns; Phenomenex, USA). The sample volume was 10 μL. The mobile phase (MP) consisted of a 50 μM solution of potassium dihydrogen phosphate, adjusted with phosphoric acid to pH 2.4 (buffer) and acetonitrile (volume ratio 83.5: 16.5) at a MP rate of 1.0 mL/min. Detection was carried out using a UV detector at 304 nm (Waters 2489, Waters Associates, Milford, MA, USA). The calibration graphs were linear ($r^2 \geq 0.99$) in the range of meropenem concentrations from 1.0 to 1000.0 μg/mL. The relative standard deviations (n = 5) for the concentrations of meropenem 1000.0, 50.0, 5.0 and 1.0 μg/mL were 1.0, 1.2, 2.7, 5.5%. The lower limit of the quantitative determination of meropenem

was 1.0 μg/mL. The limit of detection was 0.3 μg/mL. Representative chromatograms of the base (CSMHB broth without meropenem), meropenem standard and experimental meropenem detection are presented in **S1–S4 Figs**.

**The procedure of sample preparation.** A 150 μL sample of broth with antibiotic was placed in a 1.5 mL Eppendorf plastic tube, 150 μL of acetonitrile were added, shaken for 1 min, centrifuged for 5 min at 13000 rpm at 5˚C. A 150 μl sample of supernatant was placed in 1.5 mL Eppendorf plastic tubes, 300 μL of buffer was added, shaken for 1 min, and centrifuged for 5 min at 13000 rpm at 5˚C. The supernatant was analyzed by HPLC.

**Quantitation of the antimicrobial effect.** Bacteria-containing medium from the hollow-fiber bioreactor was sampled to determine bacterial and antibiotic concentrations throughout the observation period. For bacterial enumeration, samples (100 μL) were serially diluted as appropriate and 100 μL were plated onto tryptic soy agar (Becton Dickinson, USA) plates, which were incubated at 37˚C for 24 h. If was necessary to plate samples without dilution, to eliminate the antibiotic carryover effect, the 1 mL sample was centrifuged at $3000 \times g$ for 20 min [29]. The pellet was then resuspended in the normal saline to the original pre-centrifuge volume. A 100 μL sample of this suspension was then subcultured on agar plates. The lower limit of accurate detection was 2.3 log CFU/ml (equivalent to 20 colonies per plate).

To determine the time course of meropenem-resistant bacterial concentrations, the samples were plated on Mueller–Hinton agar (Becton Dickinson, USA) plates with meropenem concentration equal to 2×, 4×, 8× and 16×MIC. If the sample dilution was not applied and carryover was expected, the above described technique to avoid carryover was used. The inoculated plates were incubated for 24–48 h at 37˚C and screened visually for growth. The lower limit of detection was 1 log CFU/ml (equivalent to at least one colony per plate).

To quantify the antimicrobial effect of meropenem, an integral parameter, ABBC [30] determined as the area between the control growth and time-kill curves was calculated. The interrelation of ABBC with the antibacterial effect is direct: the greater the effect, the higher the ABBC. Time courses of bacterial counts resistant to meropenem (for each meropenem resistance level, from 2× to 16×MIC) in pharmacodynamic experiments were characterized by the integral parameter similar to ABBC, but for meropenem-resistant organisms, $ABBC_R$— area between the curves of meropenem-resistant cells obtained with and without antibiotic exposure. The interrelation of $ABBC_R$ with bacterial resistance is direct: the greater the resistance, the higher the $ABBC_R$. Both integral parameters were calculated from time zero to 120 h after the start of treatment. To assess changes in *K. pneumoniae* susceptibility to meropenem after antibiotic exposure, the ratio of MIC for total bacterial population at the end of the observation ($MIC_{FIN}$) to its initial value ($MIC_{INIT}$), $MIC_{FIN}/MIC_{INIT}$, was calculated.

**Statistical analysis.** The reported MIC and MPC data were obtained by calculation of the respective modal values. In pharmacodynamic and growth control experiments, bacterial count data were calculated as arithmetic mean ± standard deviations for three replicate experiments. Based on these data, kinetic growth and time-kill curves were constructed. Assuming that the coefficient of variation for logCFU/ml data was less than 10%, to facilitate the Fig viewing we did not place data point error bars in order to not interfere with the kinetic curves.

The ABBC versus DOSE/MIC curve was fitted by the sigmoid function:

$$Y = a / \{1 + \exp[-(x - x0)/b]\} \tag{Eq1}$$

where $Y$ is ABBC; x is log DOSE/MIC; a is maximal values of the antimicrobial effect; $x0$ is $x$ corresponding to $a/2$; $b$ is a parameter reflecting sigmoidicity.

A Gaussian function was used to fit the $ABBC_R$ versus DOSE/MIC data sets:

$$Y = a \times \exp\{-0.5 \times [(x - x0)/b]^2\} \tag{Eq2}$$

where $Y$ is $ABBC_R$; $x$ is log DOSE/MIC; $x0$ is log DOSE/MIC that corresponds to the maximal value of $Y$; $a$ and $b$ are parameters.

The $MIC_{FIN}/MIC_{INIT}$ versus DOSE/MIC data were fitted by Gaussian function:

$$Y = Y0 + a \times \exp\{-0.5 \times [(x-x0)/b]^2)\} \qquad (Eq3)$$

where $Y$ is $ABBC_R$; $Y0$ is minimal value of $Y$; $x$ is log DOSE/MIC; $x0$ is log DOSE/MIC that corresponds to the maximal value of $Y$; $a$ and $b$ are parameters.

All calculations were performed using SigmaPlot 12 software.

## Results

### Growth control experiments and baseline meropenem resistance

In growth control experiments, total bacterial counts for all *K. pneumoniae* isolates increased quickly to the maximal level. Using population analysis, varying somewhat by isolate, growth of meropenem-resistant sub-populations was observed despite the absence of meropenem (**Fig 2**). As seen in the **Fig 2**, with *K. pneumoniae* 1128 the initial level of meropenem-resistant cells was close to the detection limit. However, bacterial growth continued over time, and at the end of the observation the number of cells resistant to 2×MIC of meropenem reached 4 logCFU/ml. In comparison, the numbers of *K. pneumoniae* resistant to 16×MIC of meropenem remained at 2 logCFU/ml. With *K. pneumoniae* 1456, initial numbers of meropenem-resistant cells (2 logCFU/ml), as well as their enrichment throughout the observation period were relatively higher (up to 5.5 logCFU/ml); there was no stratification of bacterial growth according to resistance level. Isolates 1170 and 202 demonstrated similar trends regarding enrichment of meropenem resistance: modest growth of cells resistant to 2×MIC, insignificant growth of cells resistant to 4×MIC, and no growth of the most resistant sub-populations (8× and 16× MIC of meropenem). Isolate 75 demonstrated only negligible growth of cells resistant to 2×MIC on the first day of observation with subsequent elimination.

### Meropenem pharmacodynamics with total *K. pneumoniae* population

Results of *in vitro* pharmacokinetic simulations conducted in the absence of bacteria are shown in **Fig 3**. Significant agreement was found between targeted and observed meropenem pharmacokinetic profiles (coefficient of variation did not exceed 10%); observed $t_{1/2}$ was equal to 1.37 h (targeted– 1.4 h).

In pharmacodynamic simulations, meropenem was degraded under the influence of bacterial carbapenemases for all tested isolates to a greater or lesser extent (**Fig 4**).

Therefore, meropenem was not very effective against all tested *K. pneumoniae* strains, including those meropenem-susceptible isolates with meropenem MICs ranging from 0.5 to 4 mg/L. Only slight differences in meropenem effects based on initial meropenem susceptibility were observed (**Fig 5**).

The effects of meropenem expressed as the area between the bacterial curve (ABBC) parameter were plotted against the daily dose of meropenem related to the MIC of each isolate (DOSE/MIC index) (**Fig 6**); DOSE/MIC showed a good correlation with the carbapenem effect (ABBC) and was fitted with the sigmoid function with $r^2 = 0.99$.

### Meropenem pharmacodynamics with resistant sub-populations of *K. pneumoniae*

In the pharmacodynamic experiments, the growth of meropenem-resistant cells was accompanied by poor meropenem efficacy; however, this varied among isolates (**Fig 5**). In experiments

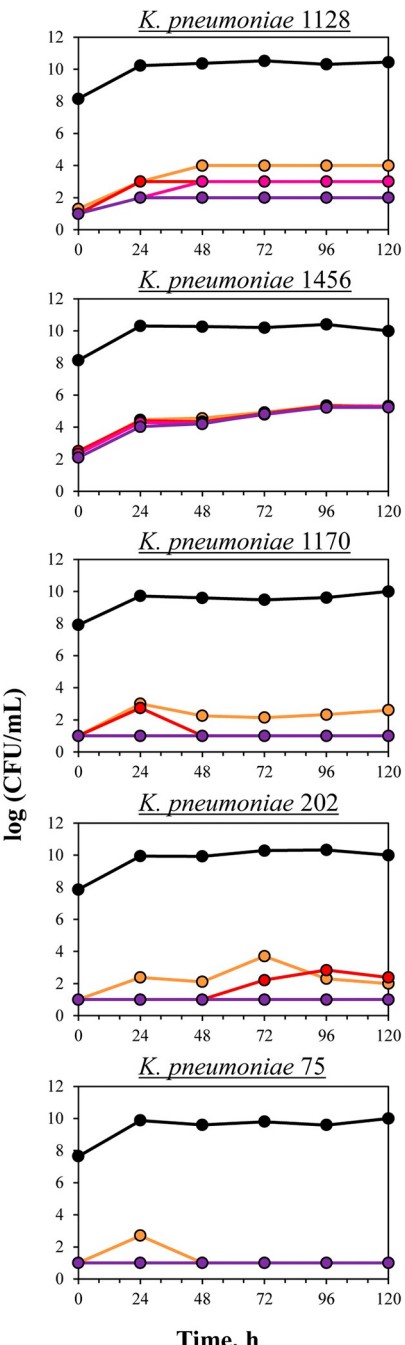

**Fig 2. Total bacterial counts and baseline meropenem resistance in control growth experiments with OXA-48-producing *K. pneumoniae*.** Time courses of the total bacterial population indicated (black circles) and meropenem-resistant (2× (orange circles), 4× (red circles), 8× (purple circles), and 16×MIC (violet circles)) sub-populations of *K. pneumoniae* in growth control experiments.

with the most susceptible *K. pneumoniae* 1128, considerable growth of resistant organisms was observed throughout the observation period: gradual growth over the first 72 hours followed by a plateau at 5–6 log CFU/ml until 120 h. However, numbers of resistant bacterial sub-populations did not reach the level of the total bacterial population. At the end of the experiment, meropenem susceptibility of the total bacterial population decreased 4-fold. Meropenem

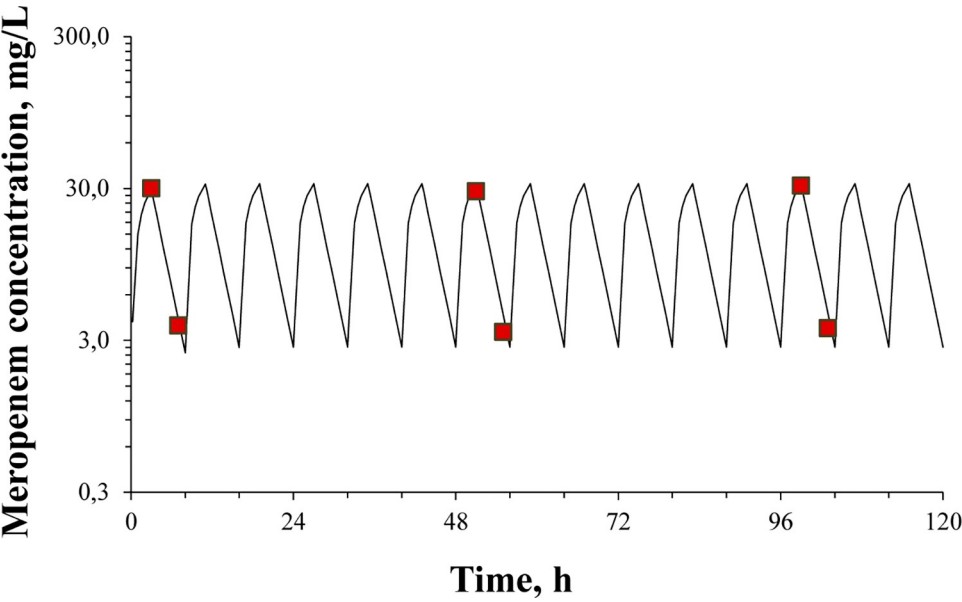

**Fig 3. *In vitro* pharmacokinetics of meropenem without bacteria.** Targeted (line) and determined (red squares) concentrations of meropenem in *in vitro* pharmacokinetic experiments.

susceptibility of *K. pneumoniae* isolates exposed to the antibiotic in pharmacodynamic experiments at the end of the observation period coupled with initial meropenem susceptibility data are shown in **Table 1**. Maximal enrichment with meropenem-resistant organisms was observed with *K. pneumoniae* 1456 with meropenem MIC of 2 mg/L; even after the 24 hours of meropenem exposure, numbers of resistant cells reached the maximal level equal to that of the total population. All bacterial cells became meropenem-resistant; this was confirmed by susceptibility data at the end of the experiment that showed a 32-fold shift in MIC. The other three *K. pneumoniae* isolates (1170, 202 and 75) exhibited low to moderate development of meropenem resistance that was inversely related to meropenem MIC: the higher the MIC, the less intensive the selection of resistant cells. Time-kill curves stratification was observed according to the level of meropenem resistance; however, the most resistant to antibiotic isolate (75) did not show noticeable development of resistance, with growth only of cells resistant to 2×MIC of meropenem. The described kinetics of growth of meropenem-resistant organisms were accompanied by consistent susceptibility changes in the total bacterial population at the end of meropenem exposure– 8-fold with 1170, 4-fold with 202 and no MIC change with isolate 75.

## DOSE/MIC relationships with the emergence of meropenem resistance

To consider the baseline enrichment of bacterial populations with resistant cells and to reflect resistance exhibited strictly under meropenem exposure, the integral parameter $ABBC_R$ was calculated. Then, "DOSE/MIC—$ABBC_R$" relationships were constructed for cells at all resistance levels (from 2× to 16×MIC); as an example, data for cells resistant to 4× meropenem MIC are shown in **Fig 7**. These relationships for *K. pneumoniae* cells resistant to 2×, 8× and 16×MIC had similar patterns (**S5 Fig**). As seen in **Fig 7**, the DOSE/MIC ratio demonstrated a good correlation with *K. pneumoniae* resistance, and a bell-shaped relationship was fitted with Gaussian function ($r^2$ 0.98). Although it was impossible to calculate an observed $T_{MSW}$, the index values based on targeted meropenem concentrations were consistent with enrichment of the bacterial population with resistant cells: the higher the $T_{MSW}$–the more intensive the development of resistance.

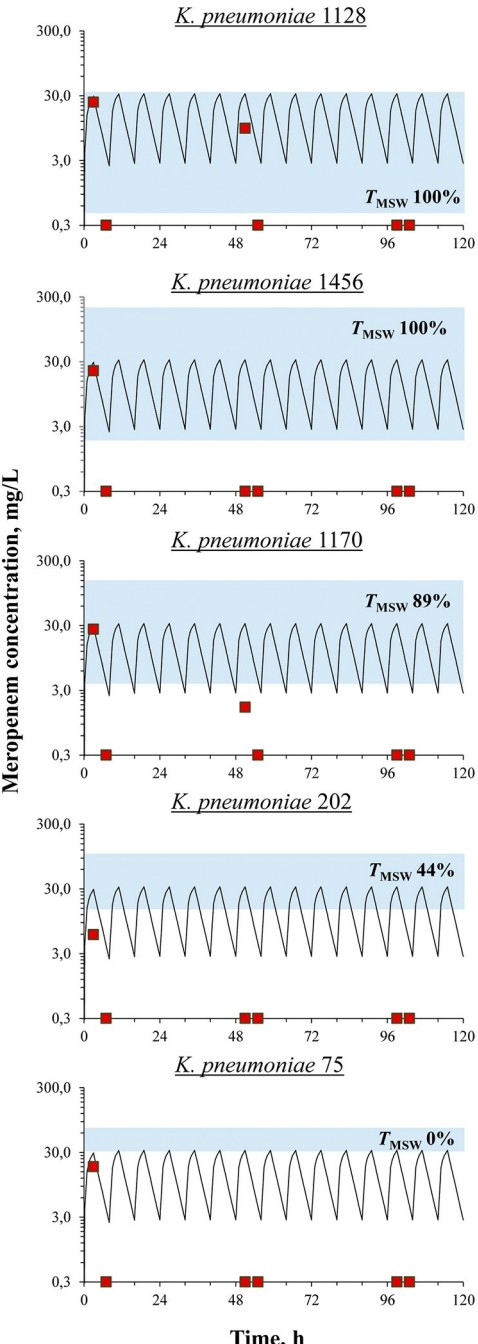

**Fig 4. *In vitro* pharmacokinetics of meropenem in the presence of bacteria.** Targeted (line) and determined (red squares) concentrations of meropenem in pharmacodynamic experiments with *K. pneumoniae* isolates. Mutant selection windows for each *K. pneumoniae* isolate are marked by shaded areas.

Similar observations could be made concerning relationships of the DOSE/MIC index and resistance expressed by the ratio of bacterial population meropenem susceptibility before and after antibiotic exposure–$MIC_{FIN}/MIC_{INIT}$ (**Fig 8**). Again, the DOSE/MIC ratio reflected the development of *K. pneumoniae* resistance, and a bell-shaped relationship was fitted by the Gaussian function (Eq 3) with a high squared correlation coefficient ($r^2$ 0.99).

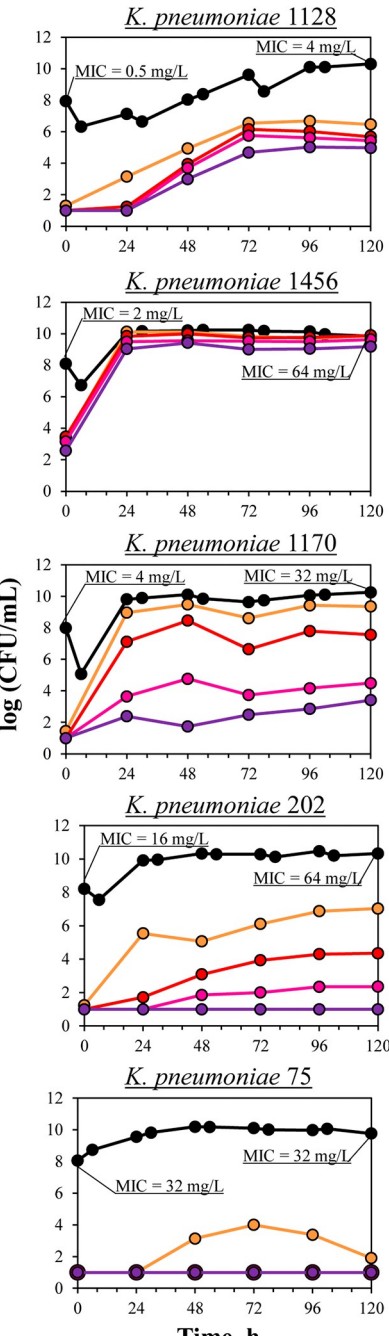

**Fig 5. Total bacterial counts and development of resistance in OXA-48-producing *K. pneumoniae* under the meropenem exposure (therapeutic regimen 2 grams every 8 hours as a 3-hours infusion).** Time courses of the total bacterial population (black circles) and meropenem-resistant (2× (orange circles), 4× (red circles), 8× (purple circles), and 16×MIC (violet circles)) sub-populations of *K. pneumoniae* in pharmacodynamic experiments.

## Discussion

In the current study we investigated the ability of meropenem to suppress the growth of meropenem-susceptible OXA-48-producing *K. pneumoniae* clinical isolates with special emphasis on resistance patterns exhibited by meropenem-exposed *K. pneumoniae*.

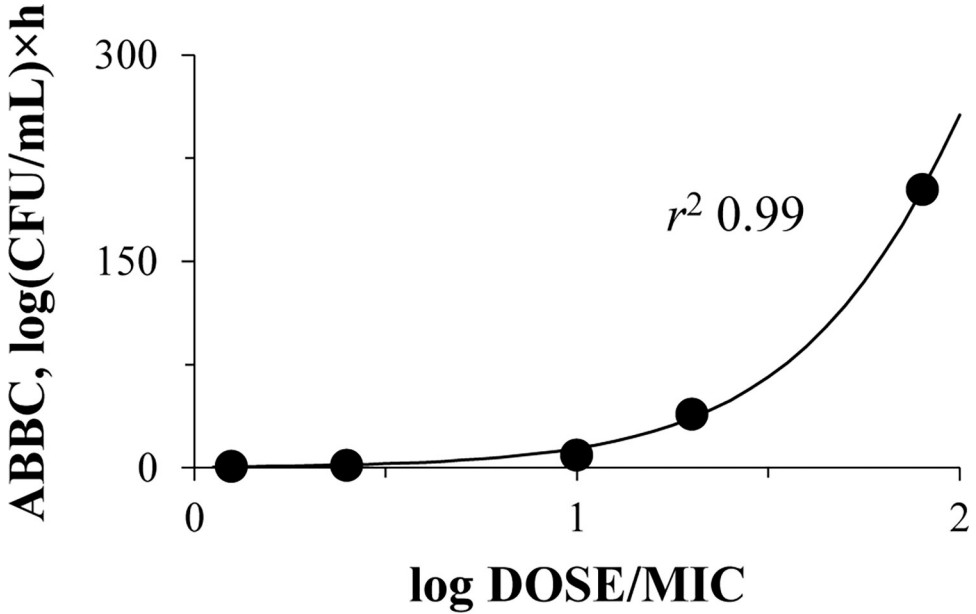

**Fig 6. Relationship between the meropenem effectiveness and its total daily dose related to the *K. pneumoniae* MIC.** DOSE/MIC relationship with ABBC fitted by Eq (1): $x0 = 2.34$, $a = 450$, $b = 0.32$.

In the current pharmacodynamic experiments, meropenem exposure was formulated to establish "concentration–response" relationships. Based on the observed antibiotic concentrations, which were noticeably lower than the desired levels with all *K. pneumoniae* isolates for most of the observation period, PK/PD indices such as time in percent of the dosing interval when antibiotic concentration is above the MIC ($T_{>MIC}$) and the ratio of the area under the concentration-time curve (AUC) to MIC (AUC/MIC), were negligible or close to zero. Therefore, it was not possible to use them for any correlations with the antimicrobial effect. Nevertheless, one can question why the development of meropenem resistance was observed in time-kill experiments, and why resistance differed among *K. pneumoniae* isolates when the antibiotic was hydrolyzed and did not work? Even though meropenem was degraded, it appears to have interacted with the microbes, contributing to the growth of meropenem resistant bacteria. The main point is that the system worked properly (without bacteria, observed meropenem concentrations were close to the targeted values, Fig 3), and desired antibiotic doses were regularly infused. In light of this we can safely propose the use of the daily meropenem dose (that was actually inserted to the system), i.e. DOSE/MIC index, for correlations with the antimicrobial effect and emergence of resistance. This approach was used for all quantitative data analyses.

Meropenem did not show any considerable effect against OXA-48-producing *K. pneumoniae*, including meropenem-susceptible isolates with MICs of 0.5–4 mg/L, even though the

**Table 1. *K. pneumoniae* susceptibility to meropenem before and after antibiotic exposure in pharmacodynamic experiments.**

| *K. pneumoniae* isolate | MIC before the meropenem exposure (MIC$_{INIT}$), mg/L | MIC after the meropenem exposure (MIC$_{FIN}$), mg/L | MIC$_{FIN}$/MIC$_{INIT}$ ratio |
|---|---|---|---|
| 1128 | 0.5 | 4 | 8 |
| 1170 | 2 | 64 | 32 |
| 1456 | 4 | 32 | 8 |
| 202 | 16 | 64 | 4 |
| 75 | 32 | 32 | 1 |

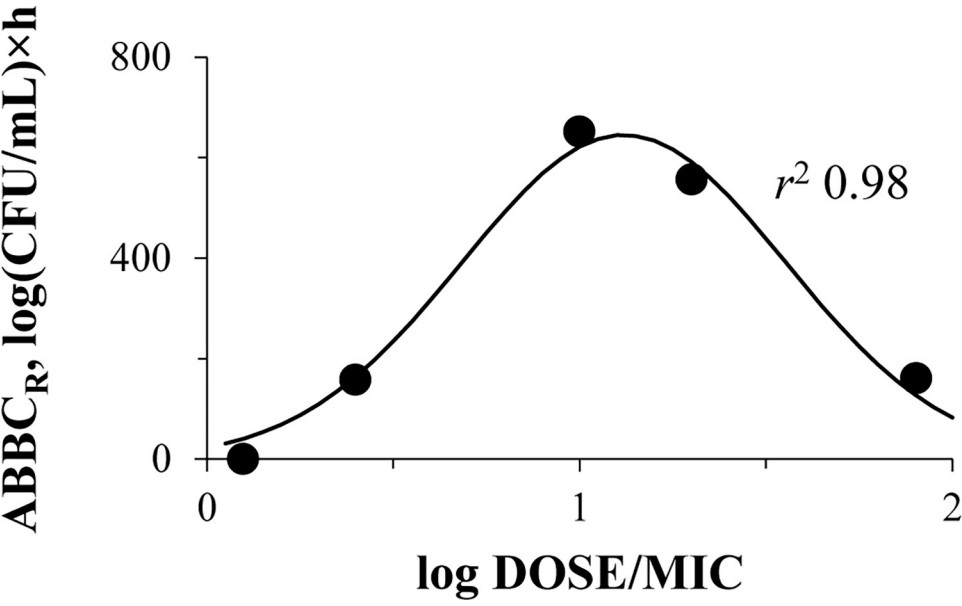

**Fig 7. Relationship between the development of meropenem resistance (expressed as parameter ABBC$_R$) in OXA-48-producing *K. pneumoniae* under antibiotic exposure and meropenem total daily dose related to MIC against the respective bacterial isolate.** DOSE/MIC relationship with ABBC$_R$ for mutants resistant to 4×MIC of meropenem; data fitted by Eq (2): $x0 = 1.12$, $a = 645$, $b = 0.43$.

simulated $T_{>\text{MIC}}$s for these isolates ranged from 89 to 100%. In a published *in vitro* study with meropenem (3, 6 and 12 h prolonged-infusions every 24 hours were simulated) and VIM-1-producing *K. pneumoniae* isolates, meropenem exposure that provided 40% $fT_{>\text{MIC}}$ was associated with a bactericidal effect [31]. In another pharmacodynamic study with KPC-

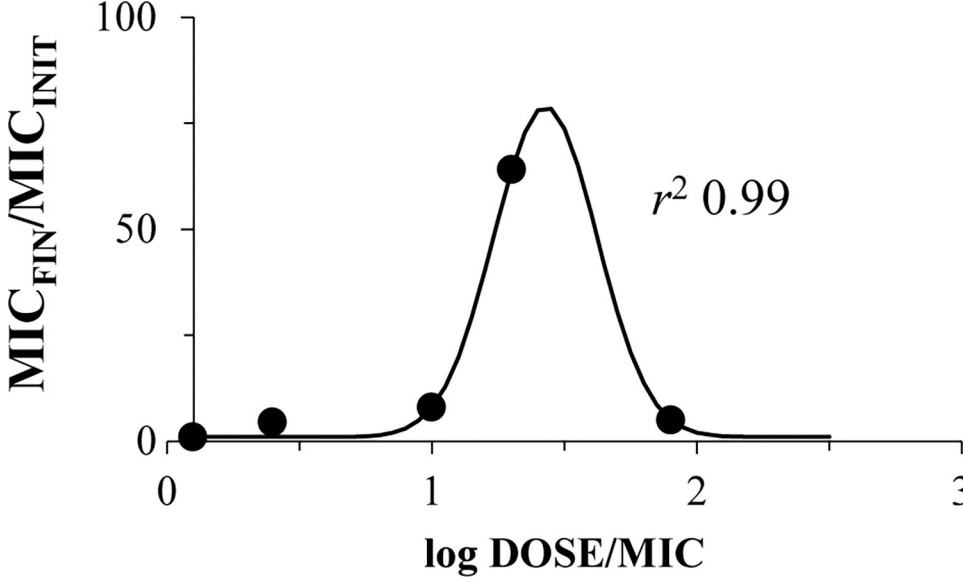

**Fig 8. Relationship between the development of meropenem resistance (expressed as MIC$_{\text{FIN}}$/MIC$_{\text{INIT}}$ ratio) in OXA-48-producing *K. pneumoniae* under antibiotic exposure and meropenem total daily dose related to MIC against the respective bacterial isolate.** DOSE/MIC relationships with MIC$_{\text{FIN}}$/MIC$_{\text{INIT}}$ ratio; data fitted by Eq (3): $Y0 = 1$, $x0 = 1.43$, $a = 78$, $b = 0.20$.

producing *K. pneumoniae* isolates and meropenem (simulated regimen—2 g q8 h as 3-h infusions), the bactericidal effect was achieved when $fT_{>\text{MIC}}$ was equal to 100% [32]. Given the results of these studies along with the current data, it can be hypothesized that meropenem effectiveness can depend on the type of carbapenemases produced by a given organism. As noted, due to meropenem hydrolysis by bacterial enzymes, we could not use observed $T_{>\text{MIC}}$s for quantitative correlations with the antimicrobial effect. For this purpose, we created the DOSE/MIC index, which permitted an excellent description of the relationship with the meropenem efficacy (Fig 6). Using this relationship, we estimate a threshold meropenem MIC equal to 0.15 mg/L that can be associated with an antibiotic bactericidal effect (3.5 log-fold decrease of bacterial counts) against OXA-48-producing *K. pneumoniae*.

The low meropenem effectiveness against susceptible isolates can be explained by the rapid development of resistance during antibiotic exposure as well as carbapenem degradation by bacterial carbapenemases. The inherent potential of the tested isolates to generate meropenem-resistant cells can be assumed from the control growth experiments. Low to moderate enrichment of the bacterial population with meropenem-resistant organisms was observed in the absence of meropenem; this baseline growth differed among the isolates. It was more evident with isolate 1456; to a lesser extent, antibiotic-resistant cells grew with isolate 1128; the other three *K. pneumoniae* isolates were characterized by weak growth of resistant organisms. At the same time, this inherent resistance was consistently lower than in the presence of meropenem (Fig 5). To reflect the exact meropenem effect on the selection of resistant organisms, the $ABBC_R$ parameter that incorporates baseline or inherent meropenem resistance was used for subsequent correlations with the DOSE/MIC index.

The use of isolates with differing meropenem susceptibility allowed us to produce a wide range of DOSE/MIC values and enabled its relationship with resistance (Figs 7 and 8). The Figs show that accurate bell-shaped relationships fitted with Gaussian function were observed. Similar bell-shaped "PK/PD index—resistance" relationships were previously reported in several studies with non-beta-lactam antibiotics with an account of the mutant selection window (MSW) hypothesis [10–17]; in this regard, beta-lactam antibiotics were not well described. Interestingly, despite meropenem degradation throughout time-kill experiments and the resultant pharmacokinetic profiles that differed significantly from targets, $T_{\text{MSW}}$ values, calculated with desired meropenem concentrations, were consistent with the enrichment of the bacterial population with resistant cells. Indirectly, this confirms that meropenem interacted with bacterial cells and influenced the enrichment of the bacterial population with resistant variants. In this light, the applicability of the MSW hypothesis to meropenem and OXA-48-producing *K. pneumoniae* can be concluded. In several studies where the development of resistance in Gram-negative bacteria to carbapenems was investigated, results cannot be compared with the current findings. Some of these studies were conducted with *Pseudomonas aeruginosa* (not with *K. pneumoniae* or *Enterobacteriaceae*), and the authors either did not specify the mechanisms of meropenem resistance or they used bacteria with alternatives to carbapenemase production mechanisms [12, 17, 33–39]. In another pharmacodynamic study with meropenem, only one *P. aeruginosa* isolate with chromosome-encoded beta-lactamase (AmpC) was used [40]; however, these beta-lactamases do not hydrolyze carbapenems [41].

Based on the "DOSE/MIC—$ABBC_R$" relationship, the meropenem threshold MIC against OXA-48-producing *K. pneumoniae* associated with the prevention of emergence of meropenem resistance is estimated at 0.15 mg/L. This MIC is significantly lower than the meropenem breakpoint MIC of 8 mg/L [42] that currently serves as a predictor of effective therapy with this antibiotic. Our findings suggest very careful prescribing of meropenem to treat *K. pneumoniae* infections when the causative isolate is confirmed as an OXA-48-carbapenemase producer; it may be preferable to use meropenem with potent carbapenemase inhibitors or in

combination with antibiotics of other classes. In addition, it would be interesting to investigate if the observed resistance patterns also are seen with *K. pneumoniae* isolates that produce carbapenemases other than OXA-48, such as KPC and NDM. The worldwide prevalence of these carbapenemases among *K. pneumoniae* clinical isolates that also possess virulence genes portends a significant pathogenetic capacity of these organisms and emphasizes the importance of such research [3].

## Conclusions

These results support (1) the low efficacy of a meropenem regimen of 2 grams every 8 hours as a 3-hour infusion against meropenem-susceptible OXA-48-producing *K. pneumoniae* isolates; the use of meropenem in combination with antibiotics of other classes and/or carbapenemase inhibitors is encouraged; (2) the threshold meropenem MIC of 0.15 mg/L against OXA-48-producing *K. pneumoniae* to achieve a bactericidal effect and to prevent the development of resistance; and (4) the applicability of the MSW hypothesis to meropenem and OXA-48-carbapenemase producing *K. pneumoniae*.

## Supporting information

**S1 Fig. Chromatogram of the base (CSMHB without meropenem).**
(TIF)

**S2 Fig. Representative chromatogram of the meropenem standard at 18.03 μg/mL.** The meropenem peak shows a characteristic retention time of 3.76 minutes.
(TIF)

**S3 Fig. Representative chromatogram showing meropenem detection (peak concentration after the first infusion) in an experimental sample obtained from pharmacodynamic simulations with *K. pneumoniae* 202.**
(TIF)

**S4 Fig. Representative chromatogram showing meropenem detection (peak concentration after the first infusion) in an experimental sample obtained from pharmacodynamic simulations with *K. pneumoniae* 1170.**
(TIF)

**S5 Fig. Relationship between the development of meropenem resistance (expressed as parameter ABBC$_R$) in OXA-48-producing *K. pneumoniae* under antibiotic exposure and meropenem total daily dose related to MIC against the respective bacterial isolate.** DOSE/MIC relationship with ABBC$_R$ for mutants resistant to 2×MIC (a), 8×MIC (b) and 16×MIC (c) of meropenem; data fitted by Eq (2).
(TIF)

## Author Contributions

**Conceptualization:** Maria V. Golikova, Vladimir A. Ageevets, Sergey V. Sidorenko, Stephen H. Zinner.

**Data curation:** Kamilla N. Alieva, Svetlana A. Dovzhenko, Mikhail B. Kobrin, Elena N. Strukova, Alisa A. Avdeeva, Ofeliia S. Sulian, Stephen H. Zinner.

**Formal analysis:** Kamilla N. Alieva, Svetlana A. Dovzhenko, Mikhail B. Kobrin, Elena N. Strukova, Alisa A. Avdeeva, Ofeliia S. Sulian.

**Funding acquisition:** Vladimir A. Ageevets.

**Investigation:** Kamilla N. Alieva, Maria V. Golikova, Svetlana A. Dovzhenko, Mikhail B. Kobrin, Elena N. Strukova, Alisa A. Avdeeva, Ofeliia S. Sulian.

**Methodology:** Maria V. Golikova, Sergey V. Sidorenko, Stephen H. Zinner.

**Project administration:** Maria V. Golikova, Vladimir A. Ageevets.

**Resources:** Vladimir A. Ageevets.

**Supervision:** Vladimir A. Ageevets, Sergey V. Sidorenko.

**Validation:** Kamilla N. Alieva, Maria V. Golikova.

**Visualization:** Kamilla N. Alieva.

**Writing – original draft:** Maria V. Golikova, Stephen H. Zinner.

**Writing – review & editing:** Kamilla N. Alieva, Maria V. Golikova, Svetlana A. Dovzhenko, Mikhail B. Kobrin, Elena N. Strukova, Vladimir A. Ageevets, Alisa A. Avdeeva, Ofeliia S. Sulian, Sergey V. Sidorenko, Stephen H. Zinner.

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
