## [Decision Letter · Decision Letter 0]

16 May 2023

PONE-D-23-09422Testing the mutant selection window hypothesis with meropenem: in vitro model study with OXA-48-producing Klebsiella pneumoniaePLOS ONE

Dear Dr. Golikova,

Thank you for submitting your manuscript to PLOS ONE. After careful consideration, we feel that it has merit but does not fully meet PLOS ONE’s publication criteria as it currently stands. Therefore, we invite you to submit a revised version of the manuscript that addresses the points raised during the review process.

We look forward to receiving your revised manuscript.

Kind regards,

Atef Oreiby, Ph. D.

Academic Editor

PLOS ONE

Journal Requirements:

"Russian Science Foundation (grant number 21-74-10090)"

"This work was supported by the Russian Science Foundation (grant number 21-74-10090)."

"Russian Science Foundation (grant number 21-74-10090)"

Additional Editor Comments:

Please address the required comments of the reviewers. It will be useful if you entered a detailed workflow figure in material and methods section.

Reviewers' comments:

Reviewer's Responses to Questions

**Comments to the Author**

1. Is the manuscript technically sound, and do the data support the conclusions?

Reviewer #1: Yes

Reviewer #2: Yes

2. Has the statistical analysis been performed appropriately and rigorously? 

Reviewer #1: Yes

Reviewer #2: Yes

3. Have the authors made all data underlying the findings in their manuscript fully available?

Reviewer #1: Yes

Reviewer #2: Yes

4. Is the manuscript presented in an intelligible fashion and written in standard English?

Reviewer #1: Yes

Reviewer #2: Yes

5. Review Comments to the Author

Reviewer #1: Dear Authors

Thank you for your manuscript submission. Your work is interesting; however, a Major Revision is needed as below:

1. Please do read and add the following papers to References section of the manuscript to have fruitful Introduction section:

Emerging multidrug-resistant bacterial pathogens "superbugs": A rising public health threat. Front Microbiol. 2023 Feb 1;14:1135614. doi: 10.3389/fmicb.2023.1135614. PMID: 36819057; PMCID: PMC9930894.

Carbapenem-Resistant Klebsiella pneumoniae: Virulence Factors, Molecular Epidemiology and Latest Updates in Treatment Options. Antibiotics (Basel). 2023 Jan 21;12(2):234. doi: 10.3390/antibiotics12020234. PMID: 36830145; PMCID: PMC9952820.

Metallo-ß-lactamases: a review. Mol Biol Rep. 2020 Aug;47(8):6281-6294. doi: 10.1007/s11033-020-05651-9. Epub 2020 Jul 11. PMID: 32654052.

2. It is recommended to add the related References associated with the used protocols in Materials and Methods section.

3. Please do add a Flow chart to Materials and Methods section to show all the procedures done in the present study.

4. It is recommended to explain about the sampling: How did you isolate the bacterial strains? Where did you isolate the bacterial strains? Which clinical samples were used to isolate the mentioned bacterial strains? Please do add the process of isolation from patients (the gender and the age range of patients) to lab.

5. Please do add the place of the study (City and Country) and the duration of the sampling and the study (from MM/YYYY to MM/YYYY)?

6. It is necessary to add an effective table to show the important Results obtained from this study.

7. Please do add the figures pertaining to HPLC section.

8. It is recommended to read and add the following paper to References section of the manuscript to have fruitful Discussion section:

Virulence factors, antibiotic resistance patterns, and molecular types of clinical isolates of Klebsiella Pneumoniae. Expert Rev Anti Infect Ther. 2022 Mar;20(3):463-472. doi: 10.1080/14787210.2022.1990040. Epub 2021 Oct 28. PMID: 34612762.

Reviewer #2: Dear Author

This is a good manuscript in the right direction to understand resistance for these bacteria and give better insight into therapeutic doses. As you have highlighted in the manuscript, it is difficult to compare your work with others since there is no standardisation of this technique for carbapenamase expressing bacteria. This is clearly going to hinder the field in making any tangible movement to better understanding of treating these infections.

It would be good if you could devise a future study that will use an ATCC characterised strain of Klebsiella that all other researchers can use as a control so as to make it possible to compare results from different labs.

Recently, David Nicolau gave some guidelines on how we can perform in vivo studies of these strains in a uniform way and I believe that this will help to compare results between labs and continents.

6. PLOS authors have the option to publish the peer review history of their article (what does this mean?). If published, this will include your full peer review and any attached files.

Reviewer #1: **Yes: **Payam BEHZADI

Reviewer #2: **Yes: **Tricia Naicker

---

## [Author Response · Author response to Decision Letter 0]

29 Jun 2023

Response to Academic Editor and Reviewers Comments

Dear Dr. Atef Oreiby,

Thank you very much for sending your and referees comments/suggestions, which were addressed in the revised manuscript. Here are our comments.

I. Your personal comment

Please address the required comments of the reviewers. It will be useful if you entered a detailed workflow figure in material and methods section.

I – Response

We included the Figure (Fig 1) displaying the experimental process flowchart comprising steps used to obtain the dataset for the analysis of “concentration – response” relationships.

II. Comments of Referee 1

Dear Authors

Thank you for your manuscript submission. Your work is interesting; however, a Major Revision is needed as below:

II.1. Please do read and add the following papers to References section of the manuscript to have fruitful Introduction section:

1. Emerging multidrug-resistant bacterial pathogens "superbugs": A rising public health threat. Front Microbiol. 2023 Feb 1;14:1135614. doi: 10.3389/fmicb.2023.1135614. PMID: 36819057; PMCID: PMC9930894.

2. Carbapenem-Resistant Klebsiella pneumoniae: Virulence Factors, Molecular Epidemiology and Latest Updates in Treatment Options. Antibiotics (Basel). 2023 Jan 21;12(2):234. doi: 10.3390/antibiotics12020234. PMID: 36830145; PMCID: PMC9952820.

3. Metallo-ß-lactamases: a review. Mol Biol Rep. 2020 Aug;47(8):6281-6294. doi: 10.1007/s11033-020-05651-9. Epub 2020 Jul 11. PMID: 32654052.

II.1. – Response

We agree with the Reviewer’s comment; the proposed papers will help to introduce readers to the subject of our research. We extended the introduction section (Lines 46-50) and included the suggested references in the Reference list [Refs 1-4 in the revised manuscript].

II.2. It is recommended to add the related References associated with the used protocols in Materials and Methods section.

II.2 – Response

We wish to clarify that all study protocols were referenced in the respective publications in the original manuscript: MIC testing procedure [ref 21], MPC testing procedure [ref 22, 23], in vitro dynamic model operational and quantitation of the antimicrobial effect procedures [ref 24-26].

II.3. Please do add a Flow chart to Materials and Methods section to show all the procedures done in the present study.

II.3. – Response

We are grateful to the Reviewer’s comment to include a flowchart to the manuscript; it will help readers better understand the study process. Accordingly, we added the Figure (Fig 1) to the Materials and methods section with the flowchart describing the steps used to prepare the dataset for further analysis.

II.4. It is recommended to explain about the sampling: How did you isolate the bacterial strains? Where did you isolate the bacterial strains? Which clinical samples were used to isolate the mentioned bacterial strains? Please do add the process of isolation from patients (the gender and the age range of patients) to lab.

II.4. – Response

According to the Reviewer’s comment we added detailed information about K. pneumoniae strains used in the study (Materials and methods section; Antimicrobial agent, bacterial isolates and susceptibility testing subsection). 

The sources of the isolated K. pneumoniae strains were: phlegmon, blood, wound and BAL. K. pneumoniae strains were identified through standard microbial and biochemical assays. The respective K. pneumoniae collection was stored at -80 ⁰C. The carbapenemase production by K. pneumoniae isolates was detected by PCR. OXA-48-positive isolates with meropenem MICs in range from 0.5 to 32 mg/L were selected for the study. Detailed clinical and demographic data of the ICU patients are not available.

II.5. Please do add the place of the study (City and Country) and the duration of the sampling and the study (from MM/YYYY to MM/YYYY)?

II.5. – Response

According to the Reviewer’s comment we provided information about place of isolation of bacterial strains and the duration of sampling (Materials and methods section; Antimicrobial agent, bacterial isolates and susceptibility testing subsection):

Clinical specimens from ICU patients in hospitals located in Moscow and Saint Petersburg (Russia) were collected from 2015 to 2019.

II.6. It is necessary to add an effective table to show the important Results obtained from this study.

II.6. – Response

We thank the Reviewer’s for suggesting the addition of a table to present important results obtained in the study. To improve the perception of the decline in bacterial susceptibility during meropenem exposure, we included these data in the Table (Table 1 in the results section). As for other data obtained in the study, the most important include the antibacterial effect of meropenem and resistance development (values of the ABBC and ABBCR integral parameters) that were used for further “concentration – response” relationship analyses. However, we do not think the display of ABBC and ABBCR values would be particularly informative and we prefer not to add these data to the table.

II.7. Please do add the figures pertaining to HPLC section.

II.7. – Response

According to the Reviewer’s comment we added the representative figures with chromatograms showing the base peaks in CSMHB (S1 Fig), detection of the meropenem standard (S2 Fig) and meropenem detection (peak concentration after the first infusion) in experimental samples with two K. pneumoniae strains, 202 (S3 Fig) and 1170 (S4 Fig).

II.8. It is recommended to read and add the following paper to References section of the manuscript to have fruitful Discussion section:

1. Virulence factors, antibiotic resistance patterns, and molecular types of clinical isolates of Klebsiella Pneumoniae. Expert Rev Anti Infect Ther. 2022 Mar;20(3):463-472. doi: 10.1080/14787210.2022.1990040. Epub 2021 Oct 28. PMID: 34612762.

II.8. – Response

According to the Reviewer’s comment we added this paper to the Reference list as well as new sentences in the introduction and discussion sections about K. pneumoniae resistance patterns among the clinical isolates (Lines 404-409 in the revised manuscript).

III. Comments of Referee 2

III.1. Dear Author

This is a good manuscript in the right direction to understand resistance for these bacteria and give better insight into therapeutic doses. As you have highlighted in the manuscript, it is difficult to compare your work with others since there is no standardisation of this technique for carbapenamase expressing bacteria. This is clearly going to hinder the field in making any tangible movement to better understanding of treating these infections.

III.1. – Response

We thank the Reviewer for this comment.

First, we would explain the technique of in vitro pharmacodynamic studies, operational procedure and viable bacteria enumeration, not influenced by the bacterial species and their ability to produce carbapenemases. At the same time, we are aware that more detailed information about lab equipment, media and manufacturers might be helpful for other researchers to compare results from different labs. To address the Reviewer’s comment, we added clarifying information about equipment used to mimic meropenem pharmacokinetics in the in vitro dynamic model and perform antibiotic dosing and sampling, and we indicate the manufacturer of media used in all testing procedures and pharmacodynamic simulations (Lines 90-91, 98, 111-113, 117-118). This is in addition to data that were specified in the original manuscript (hollow-fiber cartridge source and its model, the fiber’s material, CFU data, mimicked meropenem dose, half-life and CMAX); we hope this information will be useful for readers.

III.2. It would be good if you could devise a future study that will use an ATCC characterised strain of Klebsiella that all other researchers can use as a control so as to make it possible to compare results from different labs. Recently, David Nicolau gave some guidelines on how we can perform in vivo studies of these strains in a uniform way and I believe that this will help to compare results between labs and continents.

III.2. – Response

We absolutely agree with the Reviewer’s comment that the use of well-described bacterial strains is very important to compare study results with others. We definitely will take note at the Reviewer’s advice to use ATCC characterized bacterial strains in future studies. We support the ability to increase the reproducibility of results obtained in different labs.

We hope the revised manuscript will be acceptable for publication. Thank you for your consideration.

Sincerely, 

Maria V. Golikova

---

## [Editor Report · Decision Letter 1]

2 Jul 2023

Testing the mutant selection window hypothesis with meropenem: in vitro model study with OXA-48-producing Klebsiella pneumoniae

PONE-D-23-09422R1

Dear Dr. Golikova,

We’re pleased to inform you that your manuscript has been judged scientifically suitable for publication and will be formally accepted for publication once it meets all outstanding technical requirements.

Kind regards,

Atef Oreiby, Ph. D.

Academic Editor

PLOS ONE

---

## [Editor Report · Acceptance letter]

27 Jul 2023

PONE-D-23-09422R1 

Testing the mutant selection window hypothesis with meropenem: *in vitro* model study with OXA-48-producing *Klebsiella pneumoniae*

Dear Dr. Golikova:

I'm pleased to inform you that your manuscript has been deemed suitable for publication in PLOS ONE. Congratulations! Your manuscript is now with our production department. 

Kind regards, 

on behalf of

Dr. Atef Oreiby 

Academic Editor

PLOS ONE